# Using MALDI-FTICR-MS Imaging to Track Low-Molecular-Weight Aromatic Derivatives of Fungal Decayed Wood

**DOI:** 10.3390/jof7080609

**Published:** 2021-07-28

**Authors:** Dušan Veličković, Mowei Zhou, Jonathan S. Schilling, Jiwei Zhang

**Affiliations:** 1Environmental Molecular Sciences Laboratory, Pacific Northwest National Laboratory, Richland, WA 99354, USA; dusan.velickovic@pnnl.gov (D.V.); mowei.zhou@pnnl.gov (M.Z.); 2Department of Plant and Microbial Biology, University of Minnesota, Saint Paul, MN 55108, USA; 3Department of Bioproducts and Biosystems Engineering, University of Minnesota, Saint Paul, MN 55108, USA

**Keywords:** low-molecular aromatics, MALDI imaging, FTICR-MS, brown rot fungus, *Rhodonia placenta*, wood decomposition

## Abstract

Low-molecular-weight (LMW) aromatics are crucial in meditating fungal processes for plant biomass decomposition. Some LMW compounds are employed as electron donors for oxidative degradation in brown rot (BR), an efficient wood-degrading strategy in fungi that selectively degrades carbohydrates but leaves modified lignins. Previous understandings of LMW aromatics were primarily based on “bulk extraction”, an approach that cannot fully reflect their real-time functions during BR. Here, we applied an optimized molecular imaging method that combines matrix-assisted laser desorption ionization (MALDI) with Fourier-transform ion cyclotron resonance mass spectrometry (FTICR-MS) to directly measure the temporal profiles of BR aromatics as *Rhodonia placenta* decayed a wood wafer. We found that some phenolics were pre-existing in wood, while some (e.g., catechin-methyl ether and dihydroxy-dimethoxyflavan) were generated immediately after fungal activity. These pinpointed aromatics might be recruited to drive early BR oxidative mechanisms by generating Fenton reagents, Fe^2+^ and H_2_O_2_. As BR progressed, ligninolytic products were accumulated and then modified into various aromatic derivatives, confirming that *R. placenta* depolymerizes lignin. Together, this work confirms aromatic patterns that have been implicated in BR fungi, and it demonstrates the use of MALDI-FTICR-MS imaging as a new approach to monitor the temporal changes of LMW aromatics during wood degradation.

## 1. Introduction

Low-molecular-weight (LMW) aromatics play pivotal roles during fungal wood decomposition, acting as diffusible agents and electron carriers to mediate lignocellulose-depolymerizing reactions [1]. For example, hydroquinones can be synthesized by wood decay fungi and diffuse into the plant cell wall S2 secondary xylem layer, driving the localized production of reactive oxygen species (ROS; e.g., HO, HOO, and ROO) for breaking down wood lignocellulose [1,2,3]. Some phenolic molecules can work as electron donors for lytic polysaccharide monooxygenase (LPMO) catalyzed cellulose/hemicellulose chain oxidation [4,5]. Recently, we found evidence for a new role for LMW aromatics, as the reducing agents to scavenge oxidative radicals, thus protecting fungal hyphae and secreted enzymes from oxidative damage during brown rot wood decay [6].

Brown rot fungi represent a unique group of wood decomposer fungi that rely on highly reactive ROS species for fast wood depolymerization [7,8]. Given the risk of self-inflicted damage, ROS production would need to be restricted to the early decay stages, occurring at a safe distance away from the fungal cells and hydrolytic enzymes [8,9]. This spatiotemporal segregation can be achieved via differential gene expression but also by fungal secreted LMW aromatics that can diffuse into wood cell wall and localize, spatially, the presence of ROS. Benzoquinones are among these aromatic metabolites [2,3,10]. When in reducing formats, hydroquinones can reduce ferric iron and O_2_, respectively, to Fe^2+^ and H_2_O_2_, driving HO· generation via Fenton reaction (Fe^2+^ + H_2_O_2_ -> Fe^3+^ + HO^−^ + HO·) [11,12]. Hydrobenzoquinones including catechol, 2,5-dimethoxybenzoquinone (2,5-DMBQ), and 2,4-dimethoxybenzoquinone (2,4-DMBQ) have been identified using “bulk extraction,” and their functions in initiating Fenton oxidation of wood have been confirmed [1,2,3]. Despite of being crucial to brown rot, the intrinsic chemistry nature of the LMW aromatics is still ambiguous, and their in situ functions during brown rot have yet to be determined.

In addition to de novo synthesis, LMW aromatics can be produced from lignin degradation/modification during brown rot. Brown rot can cause extensive cleavage of inter-unit linkages of lignin polymers. It can cause the dramatic decrease of “Klason” acid-insoluble lignin in decay residues, indicating that lignin structures are depolymerized [13,14]. This depolymerization is primarily caused by oxidative attack by HO· on lignin linkages (e.g., arylglycerol-β-aryl ether, Cα-Cβ, and methoxyl bonds) that can produce a variety of aromatic products, including benzaldehydes, benzoic acids, and phenylglycerols that are readily detectable by nuclear magnetic resonance (NMR) spectroscopy [15,16]. These ligninolytic products would be further modified to soluble phenolic compounds, conferring anti-oxidizing capacities [6], or to other aryl derivatives [17,18]. This ligninolysis evidence is advancing our understanding of brown rot, and it has raised our interest in studying the LMW lignin products and elucidating their roles during advanced brown rot stages.

In general, LMW aromatics are extensively involved in brown rot, playing crucial roles for both decay initiation and continuation. Insight into temporal changes among these compounds can aid in understanding the lignocellulose-degrading mechanism of brown rot. Previous work using bulk extractions, however, cannot provide in situ information on aromatic profiles while brown rot advances [18,19,20,21]. Instead, in this work we developed a method to use matrix-assisted laser desorption ionization mass spectrometry (MALDI-MS) imaging to tract the temporal aromatic profiles on a “thin-wood-wafer” design for separating brown rot stages [8,9].

MALDI-MS has been reported as a suitable technique for profiling the chemical diversity of lignin degradation products of wood decay fungi [22,23,24]. This technology utilizes focused ultraviolet (UV) laser, in conjunction with chemical matrix (small aromatic acid or base), to ablate and ionize intact molecules from the sample surface for in situ detection, without the need of metabolites extraction. The aromatic nature of lignin degradation product resembles that of chemical matrix, making it ideal target for UV light-induced ionization. Pulsed nature of MALDI laser can be employed in imaging mode where the sample surface can be “chemically scanned,” providing spatially resolved molecular information. Combing the high mass resolution and accuracy of the Fourier transform ion cyclotron resonance (FTICR) MS system, this technology can determine the molecular formula of LMW compounds.

Here, we applied MALDI-FTICR-MS directly to wood wafers decayed by fungus *Rhodonia placenta*, a “space-for-time” design used to spatially separate the temporal brown rot stages [8,9]. After optimizing the chemical matrix to precisely monitor aromatics, this combination enabled us to temporally resolve the aromatic compounds, tracing the in situ distributions of LMW aromatics while brown rot fungus decays wood.

## 2. Materials and Methods

### 2.1. Fungal Species and Culturing Conditions

Brown rot species *Rhodonia placenta* MAD-698R (ATCC 44394; previously *Postia placenta*) was maintained on malt extract (1.5%) agar (MEA) slant at 4 °C. When needed, a mycelial edge transfer was made onto an MEA plate at 26 °C for 7–10 days.

The thin-wood-wafer microcosm design was used to culture *R. placenta* to separate different brown rot stages, as previously (Figure 1A) [9]. Soil medium containing a 1:1:1 mixture of soil, peat, and vermiculite hydrated to 40–45% (*wt/v*) moisture content was packed into a mason jar to 1/3 full, and a “feeder” strip (1/3 of a wooden tongue depressor) was then added to the soil surface to allow the fungal inoculation and mycelium mat formation on it. Sterilized aspen (*Populus* sp.) wood wafers (length × width × thickness = 60 × 25 × 2.5 mm) were propped with the 25-mm edge resting on the mycelium mat, allowing mycelial colonization from the base. Wafers were harvested when the mycelial front had progressed 50 mm up the length of a wafer, shipped to PNNL (Pacific Northwest National Laboratory, Richland, WA, USA) on dry ice, and stored at −80 °C immediately upon receipt.

### 2.2. Wood Sample Preparation

For sample preparation, frozen wafers were first dried in a desiccator for 20 min. By defining the visible hyphal front as a benchmark (0 mm; Figure 1A), wafers were cross- or longitudinal-sectioned using a pair of sharp blades, and sections were then mount onto the MALDI target plate using double-adhesive copper tape for probing the LMW aromatic compounds by MALDI-FTICR-MS (Figure 1B). Sections (thickness = 0.5 mm) collected from five individual wafers were tested, representing five bioreplicates.

### 2.3. MALDI-FTICR-MS Imaging

For MALDI-FTICR-MS imaging, DHB (2, 5-dihydroxybenzoic acid) matrix was applied over sample by robotic sprayer (HTX TM-Sprayer, HTX Technologies, Chapel Hill, NC, USA). The 40 mg/mL of DHB in 50% MeOH was sprayed with 16 passes at 50 µL/min at 80 °C, maintaining spray pressure of 10 psi (N2), a spray velocity of 1200 mm/min, and a spray nozzle distance from the sample of 40 mm. Mass spectrometry imaging was performed on a 15T MALDI-FTICR-MS (Bruker Daltonics) equipped with a SmartBeam II laser source (355 nm, 2 kHz), collecting positive ions in mass range of m/z 92–500, using a 297 ms transient, that translated to a mass resolution of R 130,000 at 400 m/z. The laser was stepped across the sample in 100 µm increments, accumulating 200 laser shots per step with frequency of 2 kHz. Imaging data were acquired using Flex Imaging (v 4.1, Bruker Daltonk).

### 2.4. Data Analysis and Discrimination of Key Aromatics

Ion imaging data processing, statistical treatment, and ion image visualization were performed using the software SCiLS^TM^ Lab (v.2021b/Release 9.01; Bruker Daltonik GmbH, Bremen, Germany, www.scils.de, 5/1/2021) [25]. Using interval width 2 mDa and relative intensity threshold >1%, peak list was created and used for downstream analyses. All ion imaging processing was performed using total ion count normalization with weak denoising and all treated individual spectra from selected regions.

To discriminate the decay stage-unique m/z ions, ROC (Receiver Operating Characteristic) values, the univariate measures to assess the discrimination quality of all m/z values through paired conditions, were calculated between paired decay stage conditions in SCiLS^TM^ Lab (v.9.01). Specifically, we set the AUC threshold values (Area Under the ROC Curve), the parameter used to estimate the discrimination quality of ROC value, as >0.6 or <0.4 to screen the upregulated or downregulated m/z ions, respectively (Appendix A). Averaged AUC values of five bioreplicate wafers were used. The changes of intensities of these screened ions (a.k.a. quantities of metabolites) along with decay stages were then clustered in heat map using Hierarchical Clustering Explorer 3.5 and by calculating Euclidean distance for complete-linkage clustering [26]. Principle component analysis (PCA) of all m/z ions characteristic for all conditions were performed with the default parameters of SCiLS^TM^ Lab (v.9.01).

Ultra-high mass accuracy of the mass spectrometer we employed enabled us to reveal exact molecular formula of imaged molecular ions. All m/z characteristic values were uploaded to metlin (https://metlin.scripps.edu, 5/1/2021) for annotation. Batch search was performed using 3 ppm accuracy and M+H and M+Na adducts. eLignin database (http://www.elignindatabase.com, 5/1/2021) [17] and reported LMW throughout literature were used to enhance the annotation confidence of aromatic compounds [1,18,27,28].

## 3. Results and Discussion

### 3.1. Temporal Changes of LMW Metabolites

Cross-sections and longitudinal sections of aspen wafers colonized by *Rhodonia placenta* were analyzed to better reflect the spatiotemporal distributions of LMW metabolites during brown rot (Figure 1A,B). A total of 3086 different m/z ions, corresponding to tentative metabolites and unknown features, were detected from all conditions. As expected, the results showed that the metabolite profiles were dramatically changed over a “time” course of brown rot and that more metabolites became detectable at later decay stages due to the lignocellulose degradation (Figure 1C) [8]. The metabolite profiles at early decay stages (0 mm) resembled that of non-decayed sections (−20 mm), although discriminate metabolites were found for brown rot initiation. Overall, metabolite changes observed in cross sections were consistent with those in longitudinal sections (Figure 1D). By using the cross sections for pairwise comparisons among non-decayed, early, mid-decay, and late decay stages, ROC analysis revealed 212 discriminate metabolites that were unique to brown rot stages. Among these discriminate ions, 40 were annotated (Appendix A). The functions of these discriminate metabolites to brown rot are discussed in the following sections.

### 3.2. Early Aromatics

The annotated discriminate metabolites (i.e., ions) were clustered according to their temporal functions during brown rot, as in Figure 2. Metabolites can be found, at detectable levels, in the sound wood even before the fungal colonization (Figure 2, clusters II and IV). This includes phosphocholine, riboluse diphosphate, and gibberelins involved in plant metabolism, and wood extractives such as arginine, flavonoids (e.g., catechins, hydroxy-trimethoxyflavan), coumarin, and phenolics. Wood extractives are secondary metabolites in plants and many of these compounds (e.g., tannins, tropolones) contribute to the natural durability of wood [29,30]. These extractives such as LMW phenolics, however, can also be employed by fungi to reduce iron and drive the generation of reactive oxygen species (e.g., HO· via Fenton reaction) for lignocellulose degradation [31]. Here, we found some phenolic metabolites exist in sound wood at relatively high levels (ion intensities > 2^13^), and this included those that were putatively annotated as acetovaillone, hydroxyacetophenone, sinapaldehyde, catechin, and guaiacylglycerol-β-guaiacyl ether. It is likely that these compounds were used to initiate the oxidative degradation of lignin polymers, and it would be worth further validating the degradative roles of these pinpointed compounds in chelating and reducing irons.

Metabolites were detected immediately at the hyphal front of *R. placenta* in wood wafers that were not detected in the non-decayed wood (Figure 2, cluster III). This included two metabolites, carnitine and spermidine, that are involved in cellular growth and energy metabolism, suggesting the active growth of fungal cells. Fungal hydroquinones such as dimethoxyhydroquinone, catechol, and variegatic acid were often linked to Fenton chemistry during early brown rot stages [1,12,32] and disappeared at later decay stages [1,2,10,12,30]. However, these metabolites were not detected from the early decay stages (0 mm) of *R. placenta* in the current study, probably, due to the inadequate detection limits of the current method.

Although no early decay-specific hydroquinones were detected, we found that the levels of several new phenols/polyphenols (e.g., catechin-methyl ether, dihydroxy-dimethoxyflavan, acetovanilone, dimethyl benzoquinones, and caffeic acid; 0 mm vs. −20 mm > 1.5-fold) were increased immediately after the fungal colonization (Figure 2). Notably, the ion intensities of catechin-methyl ether and dihydroxy-dimethoxyflavan at 0 mm increased by 7.2- and 1.9-fold, respectively, relative to non-decayed wood at −20 mm. Because phenols/polyphenols can drive the production of Fenton reactants H_2_O_2_ and Fe^2+^ [31,33], it is possible that these metabolites are used by brown rot for the early oxidation of wood polymers. It is also possible that these metabolites were used to facilitate oxidative enzymes function during early brown rot, given that phenols/polyphenols can be mediators/electron donors of laccase and lytic polysaccharide monooxygenase that are primarily expressed during early brown rot [4,7,27,34].

### 3.3. Late Aromatics

Low-molecular-weight lignin metabolites were clearly accumulating following the oxidative attack by brown rot, including phenylpropanoids, benzaldehydes, and dimeric aryl-O-aryl ethers. The phenylpropanoids such as coniferyl alcohol, coniferyl aldehyde, sinapaldehyde that were generated by HO· attacks on the β-O-4 bonds, as detected by Yelle et al. (2011) [15], showed significantly increased levels in the mid-decay stages (20 mm). As brown rot progressed, these phenylpropanoids would be further oxidized and modified to their phenylpropanoic acid formats, including coumaric acid, sinapinic acid, and cinnamic acid (cluster I in Figure 2), although we do not know the specific enzymatic reactions involved in these conversions. On the other hand, benzaldehydes such as syringaldehyde (SA), vanillin, and hydroxyacetophenone (HA), generated via another ligninolysis route by attacking Cα-Cβ bonds by HO· [15], were also detected during later stages of brown rot. Although benzoic acids have also been proposed as the oxidized products of this ligninolytic route, these were not detected in our work. Together, in agreement with a combination of others’ work but using a single, different tool [15,35], our detection of LMW lignin metabolites confirmed that brown rot process can cause significant depolymerization, demethylation, oxidation, and hydroxylation of lignin polymers.

Interestingly, we observed that hydroxy-methylbenzalpyruvate, methylmuconolactone, and carboxybenzaldehyde (CBA)/phenylglyoxalic acid (PGA) were produced during later stages of brown rot. These metabolites are all products of microbial metabolism of aromatic compounds. Muconolactones are known as intermediates in the metabolism of lignin-related aromatics via protocatechuate metabolic pathway [36]. This might imply that brown rot fungus may still be able to catabolize the degraded lignins, but with a limited rate due to the loss of complete intracellular pathways for lignin catabolism when evolved from white rot ancestor [37].

## 4. Conclusions

The aromatic nature of metabolites involved in lignin degradation makes them ideal targets for UV light-induced ionization by MALDI. Herein, we demonstrated the use of this technique for in situ molecular imaging on wood substrates degraded by a brown rot fungus, monitoring the time sequence of associate fungal LMW aromatics during wood decomposition. The key aromatics that were identified by MALDI-FTICR-MS for both early and advanced stages of brown rot were highlighted in Figure 3, pinpointing targets for further research on understanding brown rot metabolites. Although some aromatics that were thought as important to early oxidative decay of brown rot (e.g., dimethoxybenzoquinone and other catechols) were not detected by the current procedure, we anticipate that further optimizations based on the current design and different MALDI matrices, mass analyzers and orthogonal imaging measurements will more adequately capture the richness of chemistry. Together, we envision that the application of MALDI-FTICR-MS for tracking the “real-time” aromatics can advance our understanding of fungal wood decomposition mechanisms.

## Figures and Tables

**Figure 1 jof-07-00609-f001:**
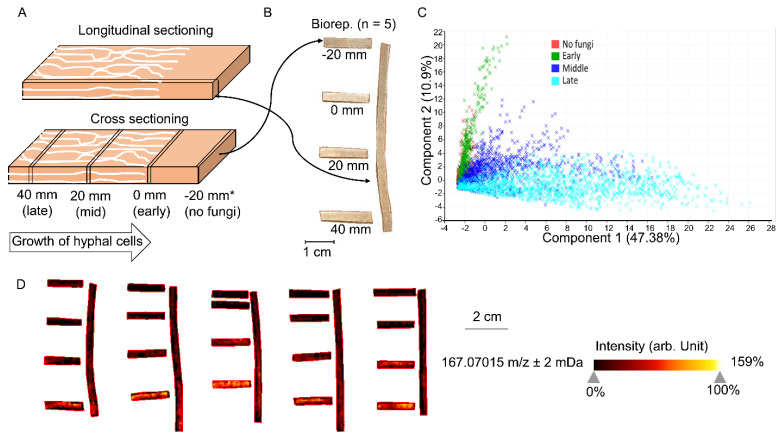
**MALDI-FTICR-MS analysis of wood wafers decayed by *Rhodonia placenta*, showing the temporal changes of LMW metabolites**. (**A**) The longitudinal section parallel with fungal growth direction and four cross-section slices at different distances from the hyphal front (−20, 0, 20, 40 mm) were sampled using a pair of blades to track the metabolites from early to late decay stages. (**B**) The generated slices with thickness of 0.5 mm were then mounted onto a target plate for MALDI-FTICR-MS analyses. (**C**) PCA plot of MS data over five bioreplicates was used to show the metabolite distributions across different stages of brown rot. Each dot represents one spectrum (pixel) from the analyzed slice. (**D**) MALDI-FTICR-MS images of one representative ion 167.07015 m/z ± 2 mDa are shown as an example, to delineate the temporal changes of metabolites along with brown rot. The specific intensities of this ion in five replicates were shown in Appendix A.

**Figure 2 jof-07-00609-f002:**
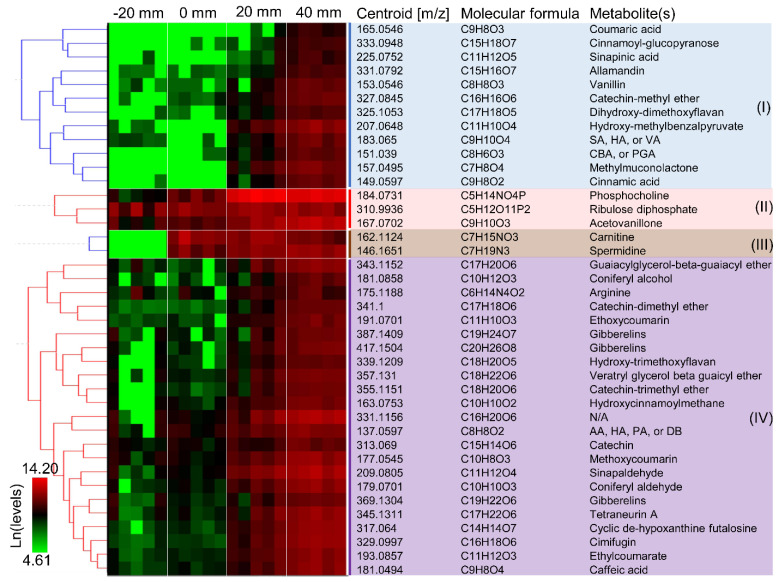
**Clustering of discriminate metabolites according to their temporal functions during brown rot.** Forty metabolites are shown with their m/z ions, molecular formula, and tentative identity. The labels −20, 0, 20, and 40 mm represent non-decayed, early, mid-decay, and late decay stages, respectively. Intensity values of five bioreplicates of metabolites were used for hierarchical clustering with HCE3.5 after ln transformation.

**Figure 3 jof-07-00609-f003:**
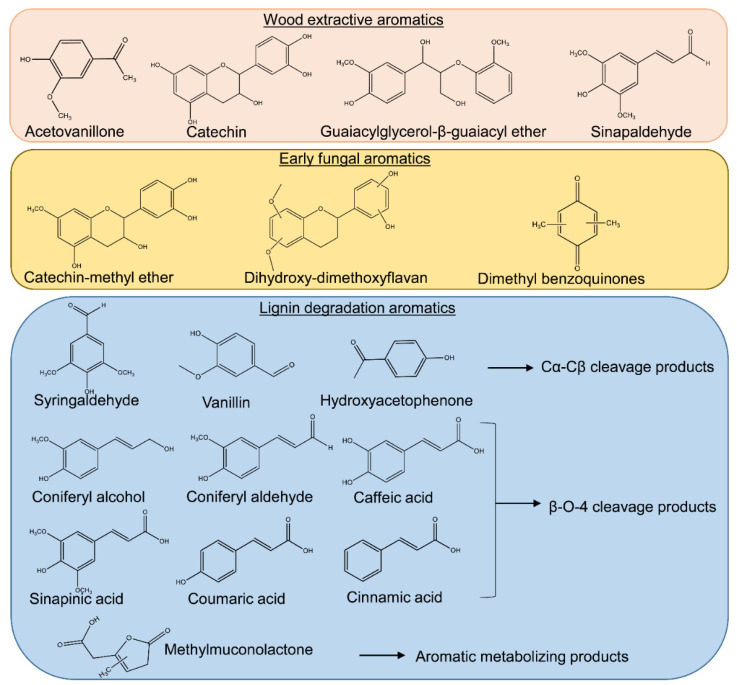
**Key aromatic metabolites detected in both brown rot initiation and continuation in R. *placenta***. Tentative aromatic identities were pinpointed by MALDI-FTICR-MS for wood extractives, early stages, and advanced stages of brown rot. The putative lignin degradative pathways and their corresponding ligninolytic products were identified according to previous studies for advanced brown rot stages [1,15,16].

## Data Availability

All the involved data have been made available along with the submission of the manuscript.

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
