# Peer review of "Using MALDI-FTICR-MS Imaging to Track Low-Molecular-Weight Aromatic Derivatives of Fungal Decayed Wood"

_jof, 2021, doi:10.3390/jof7080609_

Round 1

Reviewer 1 Report

 This is a very nice publication, and one of the rare ones that deal with instrumental methods of analysis such as MALDI in wood decay analysis. I like this, and the only requirements of this reviewer are with only minor revisions centering on the additions to be done to the references, references list and the introduction, of papers in the same line , namely on MALDI analysis of decayed wood, that are important to what these authors have done. Please do add the following two references:

  1. E.Bari, A.Pizzi, O.Schmidt, S.Amirou, M.A.Tajik-Ghanbary, M.Humar, Differentiation of fungal destructive behaviours of wood by the white-rot fungus Fomes fomentarius by MALDI-TOF mass spectrometry, J.Renew.Mater. 9 (3): 381-397 (2021) DOI:10.32604/jrm.2021.015288
  2. O.Schmidt, W.Kallow.  Differentiation of indoor wood decay fungi with MALDI-TOF mass spectrometry. Holzforschung, 59(3), 74–377 (2005). DOI 10.1515/HF.2005.062
  3. For the rest congratulations to the authors for a nice piece of research work.

Author Response

Response: Thank you for both editor's and reviewer's suggestions. We have updated the references and the relevant introduction according to the comments. Please see the changes highlighted in red in the attachment.

Reviewer 2 Report

We believe that the featured article "Using MALDI-FTICR-MS imaging to track low-molecular-weight aromatic derivatives of fungal decayed wood" is well written and should be published if minor revisions are made.

Following are those aspects that should be clarified to the reader:

  1. It would be convenient to explain figure 1 in more detail, especially for those readers interested in using this MALDI-FTICR-MS image analysis technique. For example, explain in more detail why the ion 167.07015 m / z ± 2 mDa has been chosen as representative. It would be convenient to better visualize in figure 1D the table of intensities of the 5 bioreplicas (159% arb Unit)

.

2.- Table S1 is not displayed with the 40 ions discriminated. However, we observe in figure 2 that the most representative molecular ions do appear, so in that case we would not advise including Table S1

Author Response

Response to reviewer 2's point 1: Thanks to the reviewer's comments. We have listed all the details for the MALDI-FTICR-MS image in the methods (2.2-2.4), including from sample preparation to MALDI-FTICR-MS imaging and data analyses. This information should allow others to use this technique.

The ion 167.07015 m / z ± 2 mDa was chosen as a representative for delineating the temporal changes of metabolites along with brown rot. Its intensities in 5 bioreplicates were shown in Table S1. To further clarify this, we added a sentence of "The specific intensities of this ion in five replicates were shown in Table S1." in the figure caption.

Response to reviewer 2's point 2: We are sorry to forget to upload Table S1. Given this table includes more details of the discriminate ions, including the intensities of 5 bioreplicates and the significant analysis, we believe it would be still necessary to include it in this report.

All the changes have been highlighted in red in the attached file. Please let us know if there are any questions.
